# Increased Prevalence of Elevated DHEAS in PCOS Women with Non-Classic (B or C) Phenotypes: A Retrospective Analysis in Patients Aged 20 to 29 Years

**DOI:** 10.3390/cells11203255

**Published:** 2022-10-17

**Authors:** Enrico Carmina, Rosa Alba Longo

**Affiliations:** Endocrinology Unit, University of Palermo School of Medicine, 90133 Palermo, Italy

**Keywords:** PCOS, PCOS phenotypes, DHEAS, adrenal androgens, adrenal hyperandrogenism in PCOS

## Abstract

It is well known that a subgroup of women with PCOS present an excessive adrenal androgen production, generally associated with ovarian hyperandrogenism. In the past, it has been impossible to correlate adrenal hyperandrogenism to any clinical or hormonal pattern of PCOS. However, adrenal androgens are strictly dependent on age and their blood values reduce by 40% in patients moving from their twenties to thirties. Due to this, serum DHEAS values are strongly influenced by the age distribution of studied populations. To avoid this bias, in this study we retrospectively analyzed the clinical and hormonal data of PCOS women in their twenties (age between 20 and 29 years). Data of 648 young hyperandrogenic women with PCOS were evaluated. Serum DHEAS was increased in a third (33%) of studied patients and was associated with higher values of testosterone (T) and androstenedione (A). In each phenotype, patients with high DHEAS had higher values of T and A than patients with normal DHEAS of the same phenotype. Therefore, a DHEAS increase is generally part of a generalized higher androgen production in a subgroup of PCOS patients, independently of the finding of anovulatory or ovulatory cycles or of polycystic or normal ovaries. However, our study showed some important differences between PCOS phenotypes. A lower prevalence of increased DHEAS in A phenotype PCOS patients who generally have the highest androgen levels, versus non-classic (B or C) PCOS phenotypes, was observed. It was also found that patients with A phenotype PCOS present significantly lower BMI and serum insulin than patients with normal DHEAS of the same phenotype while, in patients with the B or C phenotype, the opposite occurs. We conclude that adrenal hyperandrogenism is more common in patients with non-classic (B and C) phenotypes of PCOS and is generally part of a generalized higher production of androgens in a subgroup of PCOS patients. However, other factors may increase the adrenal androgen production and influence the clinical expression of the syndrome. More studies in large, selected for age, populations of PCOS women with different phenotypes are needed.

## 1. Introduction

It is well known that a subgroup of women with PCOS present an excessive adrenal androgen production, generally associated with ovarian hyperandrogenism [1,2,3]. In the past, it was impossible to correlate the finding of adrenal hyperandrogenism to any specific clinical or hormonal finding of the syndrome. However, most previous studies were performed in PCOS women diagnosed by NIH criteria (anovulatory hyperandrogenic patients) [1,2,3].

PCOS is a very heterogeneous disorder and the introduction of Rotterdam diagnostic criteria (chronic anovulation, hyperandrogenism and polycystic ovaries, two out of three) has partially expressed the possible clinical and hormonal characters of patients affected by PCOS [4,5]. In fact, patients with PCOS may be anovulatory and hyperandrogenic with polycystic ovaries (phenotype A), but also may have normal ovaries (phenotype B) or be ovulatory (phenotype C) or also have normal circulating androgens and no clinical signs of androgen excess (phenotype D) [4,5]. In the past, in a group of unselected women with PCOS, we did not find any difference in adrenal androgen prevalence between the different phenotypes [5,6,7].

However, adrenal androgens are strictly dependent on age [8] and several years ago Labrie and al. showed that adrenal androgen secretion reduces by 40% in women aged thirty years compared to women aged 20 years [9]. A similar decline of serum DHEAS has been found in hyperandrogenic women [10] and in PCOS [11]. Due to this, in unselected populations of women with PCOS, results may be strongly influenced by the distribution of the age of studied patients. Our previous studies included patients with age between 18 and 40 years [5,7] but a negative correlation between age and serum DHEAS was always present, suggesting that differences in age between patients influence the results and may hide the real distribution of adrenal hyperandrogenism. To avoid this bias, we conducted some preliminary studies, but in patients aged 18 to 35 years a significant negative correlation between age and serum DHEAS was also present. Finally, restricting the study only to patients in their twenties (age between 20 and 29 years), no correlations between age and serum DHEAS were present.

In this study, we report the distribution of adrenal hyperandrogenism (assessed by serum DHEAS values) in a population of PCOS women selected with an age between 20 and 29 years. Possible correlations of increased serum DHEAS values with some clinical, hormonal and metabolic characters of the disorder were evaluated, too.

## 2. Materials and Methods

Between January 2012 and July 2020, 954 consecutive patients with a diagnosis of PCOS and clinical or biochemical hyperandrogenism were studied. All these patients were referred because of hyperandrogenism and/or menstrual disorders. Most studied patients were included in our previous studies regarding PCOS phenotypes [8,12]. The data of 648 young adult PCOS women in their twenties (age range 20–29, mean age 24.3 ± 2.8 years) were retrospectively analyzed and reported in this study.

The diagnosis of PCOS was based on Rotterdam criteria, two out of three of the following criteria: chronic anovulation, clinical or biologic hyperandrogenism and/or polycystic ovaries on ultrasound, after the exclusion of other medical disorders [4].

Phenotype A PCOS was diagnosed with the finding in the same patient of all three characters of the syndrome (chronic anovulation, hyperandrogenism and polycystic ovaries). Phenotype B PCOS was diagnosed in patients presenting hyperandrogenism and chronic anovulation but no polycystic ovaries while phenotype C PCOS was diagnosed in patients presenting hyperandrogenism and polycystic ovaries but ovulatory cycles [5,8,12]. No patient with phenotype D (normo-androgenic) was included in this study. Phenotype A PCOS was present in 336 patients (51.9%), phenotype B in 48 patients (7.4%), phenotype C in 264 patients (40.7% of PCOS patients). Mean age in the three groups of patients was similar (phenotype A: 24.4 ± 2.9 years, phenotype B: 24 ± 3 years, phenotype C: 24.2 ± 2.8 years).

Normal menses were defined as cycles lasting 25–34 days. Height and weight were recorded, and BMI was calculated as kg/m^2^.

Clinical hyperandrogenism was defined as the presence of hirsutism. Hirsutism was assessed by Ferriman–Gallwey–Lorenzo scores [13], and patients with scores higher than 6 were considered hirsute. Adult acne and female pattern hair loss were not considered a sign of hyperandrogenism if androgen levels were normal [14,15].

Serum levels of luteinizing hormone (LH), follicle stimulating hormone (FSH), estradiol, total testosterone (T), androstenedione (A), dehydroepiandrosterone sulfate (DHEAS), 17-hydroxy-progesterone (17OHP) and anti-Mullerian hormone (AMH) were determined on days 3–5 of the cycle. In non-menstruating women, blood samples were obtained after withdrawal bleeding after progestogen administration. Normally menstruating patients had serum progesterone measured on days 21–22 of the cycle.

Anovulation was defined as serum progesterone < 3 ng/mL (<9.54 nmol/L). In patients with normal menses, at least two consecutive menstrual cycles were studied and a finding of low levels of serum progesterone (<3 ng/mL) in both cycles indicated the presence of chronic anovulation.

Biochemical hyperandrogenism was defined as serum testosterone > 55 ng/dL and/or serum DHEAS higher than 3 mcg/mL (>7.8 mmol/L). Steroid hormones were measured by specific RIAs using previously described methods [16]. In all patients, serum 17OH progesterone values were determined to exclude the existence of a non-classic congenital adrenal hyperplasia [17]. In some patients, because of clinical suspicion, urinary free cortisol and serum prolactin and TSH were measured by commercial RIA methods to exclude other endocrine conditions.

For AMH measurement, samples were collected into serum tubes with gel separators and centrifuged within 5 h. AMH was measured using a previously described method [18]. The conversion of AMH in ng/mL to pmol/L requires that values be multiplied by 7.143.

LH and FSH were measured by specific RIAs using previously described methods [16].

In all assays, intra-assay and inter-assay coefficients of variation did not exceed 6% and 15%, respectively.

Transvaginal pelvic ultrasound was performed using a transducer frequency of 8–10 MHz and the presence of polycystic ovaries was established by the finding of an increased number of follicles, each of which measured 2–10 mm in diameter, and/or increased ovarian size [19].

No patient had received any medication for at least 3 months before the study, and all patients gave informed consent for this evaluation and the research protocol had obtained institutional approval from the ethical committee of our university (2012/6, 2018/23).

The various values of the women with PCOS were compared to those of eighty-five, age-matched, normal ovulatory women [12]. Age of the controls ranged from 20 to 29 years (mean age 24.2 ± 3 years). These controls were drawn from the same population and did not report complaints of hyperandrogenism or menstrual irregularities.

### Statistical Analysis

Statistical analyses were performed using Statview 5.0 (SAS Institute, Cary, NC, USA). Because several values were not normally distributed, a log transformation was necessary to obtain a normal distribution. Mann–Whitney U tests were performed to compare parameters between the PCOS groups. Analysis of variance (ANOVA) followed by Tukey tests was performed to assess differences in clinical and biochemical parameters between different phenotypes. Accuracy of parameters used to discriminate between the various phenotypes of PCOS and controls was evaluated using ROC curve analyses. Differences in reliability between different parameter values were assessed by Tukey multiple comparison tests. *p* < 0.05 was considered statistically significant. All results are reported as mean ± SD.

## 3. Results

Patients with PCOS presented significantly (*p* < 0.01) higher DHEAS levels (mean ± SD 2.8 ± 1.3 mcg/mL) than controls (1.8 ± 0.6 mcg/mL). Two hundred and twelve (33%) young PCOS patients presented increased serum values of DHEAS (>3 mcg/mL). Serum testosterone, androstenedione, LH, LH/FSH ratio and AMH were significantly (*p* < 0.01) higher in PCOS women compared with controls.

Dividing PCOS patients according to their DHEAS circulating levels, PCOS patients with increased DHEAS had significantly higher values of T and A than PCOS patients with normal DHEAS (Table 1). BMI, severity of hirsutism, finding or severity of adult acne or female pattern hair loss, LH/FSH ratio, estradiol, AMH and insulin were similar in the two subgroups of PCOS women (Table 1).

No correlations of serum DHEAS with age, clinical (body weight, severity of hirsutism, finding or severity of adult acne or female pattern hair loss) or hormonal (serum LH, FSH, LH/FSH ratio, AMH, estradiol) patterns were found except for a positive correlation between serum DHEAS with serum total testosterone (*p* < 0.01) and with serum androstenedione (*p* < 0.01).

Circulating values of serum androgens in the three hyperandrogenic phenotypes of women with PCOS were compared (Table 2). Total T and A were significantly (*p* < 0.01) higher in PCOS patients with A and B phenotypes compared to PCOS women with the C phenotype. Instead, mean serum DHEAS was significantly higher (3 ± 1.4 mcg/mL, *p* < 0.01) in phenotype C patients than in phenotype A (2.6 ± 1.4 mcg/mL) (Table 1). Forty-one percent of PCOS patients with the C phenotype and 42% of PCOS patients with the B phenotype presented increased serum DHEAS compared to 25 % of the A phenotype PCOS patients.

In single phenotypes of PCOS women, patients with increased DHEAS and patients with normal DHEAS were compared. As reported in Table 3, patients of each phenotype with increased DHEAS had higher T and A circulating values than PCOS patients of the same phenotype with normal DHEAS. Interestingly, PCOS patients of the A phenotype with increased DHEAS had lower BMI and insulin levels than patients of the same phenotype with normal DHEAS values. On the contrary, patients of the B and C phenotypes had higher BMI and insulin levels than patients of the same phenotype with normal DHEAS. Other clinical (age, severity of hirsutism, finding or severity of adult acne or female pattern hair loss) and hormonal (LH/FSH ratio, estradiol, AMH) patterns were not different in patients with high or normal DHEAS of the same phenotype. Similarly, in single phenotypes of PCOS patients, no correlations of serum DHEAS with age, clinical (body weight, severity of hirsutism, finding or severity of adult acne or female pattern hair loss), hormonal (serum LH, FSH, LH/FSH ratio, AMH, estradiol) patterns were found except for a positive correlation between serum DHEAS with serum total testosterone (*p* < 0.01) and with serum androstenedione (*p* < 0.01).

Increased DHEAS levels were generally associated with increased testosterone and/or androstenedione levels. Sixteen patients with phenotype C (6.2 %) and only four patients with phenotype A (1.2%) had increased DHEAS serum levels without a concomitant increase in serum T or A.

## 4. Discussion

Adrenal hyperandrogenism is common in PCOS patients and may be assessed by several methods, including assay of circulating values of DHEAS and 11-hydroxy-androstenedione (11-OH-A) [1,2,3,20,21,22]. In clinical practice, serum DHEAS is the most used marker of adrenal hyperandrogenism and, in the past, we and others have reported that about 20–40% of women with PCOS have increased circulating values of this steroid [1,2,3,6,7]. However, it has been impossible to demonstrate any difference between women with PCOS and increased DHEAS values and women with PCOS and normal DHEAS and the possible role of adrenal androgens in pathogenesis and/or clinical expression of PCOS has remained unclear [6].

Studying different PCOS phenotypes, we also did not find any difference in increased DHEAS prevalence between PCOS phenotypes [3,6] but only a positive correlation between serum DHEAS and other circulating androgens [6] and a negative correlation between serum DHEAS and serum insulin [6].

However, DHEAS circulating values are strictly dependent on age and, when moving from their twenties to thirties, in normal [8], hyperandrogenic [9] and PCOS [10] women, serum DHEAS decreases by about 40%. Due to this, studies on DHEAS prevalence and effects in populations of women unselected for a restricted age range may give biased results. In the past, we have studied PCOS women with age ranging from 18 to 40 years but, in these previous studies, serum DHEAS showed a significant negative correlation with age, suggesting that the results could be influenced by the age distribution of the studied patients.

In this retrospective study, we decided to analyze only the data of women with PCOS in their twenties. In fact, preliminary analysis of the available data suggested that studies including patients with a larger age range were always associated with a significant negative correlation between DHEAS circulating values and age of the patients. The data of a large population of young hyperandrogenic (phenotypes A, B and C) women with PCOS aged 20 to 29 years were collected. In this age-selected group of patients, no correlation between serum DHEAS and age was found, suggesting that biases linked to different ages of patients were eliminated or strongly reduced.

Our data confirm that about one third (33%) of young women with PCOS have increased circulating values of DHEAS [1,2,3]. Comparing PCOS women with high DHEAS with PCOS patients with normal DHEAS, we found significant differences only in serum levels of other androgens (T and A) while all other clinical or hormonal parameters were similar. No correlation of serum DHEAS with the clinical or hormonal pattern of PCOS was found with the notable exception of a positive correlation with other circulating serum androgens (total T and A). It suggests that adrenal hyperandrogenism is part of a more generalized increase in serum androgens and that is more important in PCOS patients with the higher androgen production.

Comparing data between patients with high versus normal DHEAS in each phenotype, it was confirmed that increased DHEAS levels are generally an expression of a generalized higher androgen secretion. In each phenotype, independently of the finding of anovulatory or ovulatory cycles or of polycystic or normal ovaries. Patients with high DHEAS had higher values of T and A than patients with normal DHEAS of the same phenotype. Consistent with this, only in a few patients (about 6% of patients with phenotype C and very few patients with phenotype A) was the increase of serum DHEAS isolated, indicating that isolated adrenal hyperandrogenism is very uncommon in PCOS.

However, our study showed some important differences between PCOS phenotypes. A lower prevalence of increased DHEAS in A phenotype PCOS patients who generally have the highest androgen levels, versus non-classic (B or C) PCOS phenotypes, was observed. In addition, mean serum DHEAS values were higher in phenotype C patients compared to phenotype A patients. These data indicate that patients with non-classic forms of PCOS (phenotype B: normal ovaries; phenotype C: normal ovulatory cycles) present a higher prevalence of adrenal hyperandrogenism than the common classic form of PCOS (phenotype A).

It was also found that patients with A phenotype PCOS having increased DHEAS present significantly lower BMI and serum insulin than patients with normal DHEAS of the same phenotype while the opposite occurs in patients with the B and C phenotypes. The meaning of these findings is unclear and more studies in carefully selected for age, large populations of PCOS women of different phenotypes are needed.

However, putting together the difference in prevalence of increased DHEAS in non-classic (B and C) PCOS phenotypes and the opposite influence in these phenotypes on BMI and serum insulin compared to the classic A phenotype may indicate that mechanisms, other than generalized higher androgen production, may operate, increase adrenal androgens and influence the clinical presentation of PCOS.

We conclude that adrenal hyperandrogenism is more common in patients with non-classic (B and C) phenotypes of PCOS and is generally part of a generalized higher production of androgens in a subgroup of PCOS patients. However, other factors may increase the adrenal androgen production and influence the clinical expression of the syndrome. More studies in large, selected for age, populations of PCOS women with different phenotypes are needed.

## Figures and Tables

**Table 1 cells-11-03255-t001:** Some clinical and hormonal data in 648 young women with polycystic ovary syndrome (PCOS), aged 20 to 29 years, divided into normal or high (>3 mcg/mL) DHEAS serum values.

	N (%)	DHEAS mg/mL	Age (Years)	BMI	LH/FSH Ratio	AMH ng/mL	T ng/dL	A ng/mL	Insulin mU/mL
High DHEAS PCOS	59 (35%)	4.3 ± 1.1 **	23.4 ± 2.9	26 ± 6 *	1.5 ± 0.9 *	7.6 ± 5 *	77 ± 27 **	4.4 ± 2.9 **	13.2 ± 6 *
Normal DHEAS PCOS	107 (65%)	2 ± 0.6	24 ± 2.9	26 ± 6 *	1.5 ± 0.8 *	8.2 ± 5 *	67 ± 28 *	3.2 ± 1.1 *	13.3 ± 7 *
Age-matched controls	85	1.8 ± 0.6	24 ± 3	22.4 ± 4	1.1 ± 0.4	2.2 ± 1	32 ± 10	1.9 ± 0.6	8 ± 3

DHEAS: Dehydroepiandrosterone sulfate; BMI: Body mass index; LH/FSH ratio: Luteinizing hormone/follicle stimulating hormone; AMH: Anti-Mullerian hormone; T: Testosterone; A: Androstenedione. ** *p* < 0.01 versus normal DHEAS PCOS patients and controls; * *p* < 0.01 versus controls.

**Table 2 cells-11-03255-t002:** Androgen circulating values in 648 women with polycystic ovary syndrome (PCOS), selected with age between 20 and 29 years and divided according to their phenotype.

	N	DHEAS mg/mL	T ng/dL	A ng/mL
Phenotype A PCOS	336	2.6 ± 1.4 *	78 ± 30 *	3.9 ± 2.3 *
Phenotype B PCOS	48	2.8 ± 1.3	76 ± 28 *	3.8 ± 1.2 *
Phenotype C PCOS	264	3.0 ± 1.4	63 ± 26	3.4 ± 1.4

* *p* < 0.01 versus phenotype C PCOS.

**Table 3 cells-11-03255-t003:** Serum androgens and some other hormonal values in patients of the three hyperandrogenic phenotypes of women with PCOS, divided according to their DHEAS serum levels.

	N (%)	DHEAS mg/mL	Total T ng/dL	A ng/mL	LH/FSH Ratio	BMI	Insulin mU/mL
Increased DHEAS A phenotype	84 (25%)	4.3 ± 1.2 *	84 ± 26 *	5 ± 2 *	1.9 ± 1	25.4 ± 4 *	13.3 ± 4
Normal DHEAS A phenotype	252 (75%)	2 ± 0.6	74 ± 23	3.4 ± 1.3	1.8 ± 1	27.6 ± 7	14.2 ± 8
Increased DHEAS B phenotype	20 (42%)	4 ± 0.5 °	89 ± 23 °	4 ± 1.3 °	1.7 ± 1	27 ± 6 °	14.9 ± 6
Normal DHEAS B phenotype	28 (59%)	1.6 ± 0.6	62 ± 32	3.2 ± 1	1.7 ± 0.6	25 ± 5	11.9 ± 8
Increased DHEAS C phenotype	108 (41%)	4.4 ± 1 ^	68 ± 28 ^	3.9 ± 1.8 ^	1 ± 0.5	25 ± 5 ^	12.6 ± 8
Normal DHEAS C phenotype	156 (59%)	2 ± 0.6	56 ± 26	3 ± 1	1.1 ± 0.5	23.6 ± 4	11.8 ± 6

DHEAS: Dehydroepiandrosterone sulfate; T: Testosterone; A: Androstenedione; BMI: Body mass index; LH/FSH ratio: Luteinizing hormone/follicle stimulating hormone. * *p* < 0.01 versus normal DHEAS A phenotype PCOS patients; ° *p* < 0.01 versus normal DHEAS B phenotype PCOS patients; ^ *p* < 0.01 versus normal DHEAS C phenotype PCOS patients.

## Data Availability

Data supporting results can be found at the office of Prof Carmina.

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
