# Peer review of "Increased Prevalence of Elevated DHEAS in PCOS Women with Non-Classic (B or C) Phenotypes: A Retrospective Analysis in Patients Aged 20 to 29 Years"

_cells, 2022, doi:10.3390/cells11203255_

Round 1

Reviewer 1 Report

 The authors have performed a good study about an interesting subject in PCOS, but there are several shortcomings in the written manuscript which need to be addressed. Some of the needed corrections are shown in the attached PDF file, please read and correct the manuscript accordingly. There are also other important points which are mentioned below:

Introduction:

- Non-sufficient. It needs more information about PCOS, pathogenesis, different mechanisms and pathogenesis of hyperandrogenism in PCOS, and the definition of phenotypes, …

Methods:

- What is the study method, setting, population, and time?

- Were androgen levels checked before recruitment of controls?

Results:

- Results section needs to be re-written, with carefully drafted tables. The present form is entirely inadequate.

- All written material should be presented in classified tables.

- The first table should demonstrate and compare demographic and biochemical characteristics of cases and controls.

- Different tables should be added to demonstrate and compare the clinical and biochemical characteristics of 3 patient groups.

- Tables need to contain statistical methods, p-values, and footnotes for abbreviations. Post-hoc analysis should be added in the footnotes.

- Several assessments mentioned in Methods, such as insulin, and HOMA-IR, and hormonal assessments were not mentioned in Results. Why were so many tests performed if not needed to analyze? Please delete unrelated tests in the Methods section.

Discussion:

-The present form of discussion is inadequate.

- Discussion needs to go through more previous studies, compare their results with these findings, also demonstrate the differences and similarities, and propose possible mechanisms.

- Several findings discussed in this part are not adequately mentioned in Results; hence, the need to re-write the Results section!

Reviewer 2 Report

The topic of the article is with significant implications for medical practice. The text is clear and easy to read. The manuscript has an excellent methodical description. The overall paper is organized well written. The methods, the overall study design, and statistical analysis are clearly described. Discussions Section presents other research findings. The literature reviews are insightful and informative.

The table is well presented and easy to read and understand.

I have only a few remarks to make:

There are no keywords defined.

The abbreviations should be defined the first time they appear in three sections: the abstract, the main text, and the first figure or table.

No ethical statement.

The results section must be more detailed.

The references must be formatted following the instructions for the authors.

Author Response

Response to reviewer 2.

We thank the reviewer for the comments.

We tried anyway to improve the paper adding some additional analysis of the data and two additional tables.

We believe that it has improved the results section and consequentely also the discussion.

Other points, like keywords, abbreviation, ethical statement were added.